# Anticancer Activities of 9-chloro-6-(piperazin-1-yl)-11H-indeno[1,2-c] quinolin-11-one (SJ10) in Glioblastoma Multiforme (GBM) Chemoradioresistant Cell Cycle-Related Oncogenic Signatures

**DOI:** 10.3390/cancers14010262

**Published:** 2022-01-05

**Authors:** Ntlotlang Mokgautsi, Yu-Cheng Kuo, Sung-Ling Tang, Feng-Cheng Liu, Shiang-Jiun Chen, Alexander T. H. Wu, Hsu-Shan Huang

**Affiliations:** 1PhD Program for Cancer Molecular Biology and Drug Discovery, College of Medical Science and Technology, Taipei Medical University and Academia Sinica, Taipei 11031, Taiwan; d621108006@tmu.edu.tw; 2Graduate Institute for Cancer Biology & Drug Discovery, College of Medical Science and Technology, Taipei Medical University, Taipei 11031, Taiwan; 3Department of Pharmacology, School of Medicine, College of Medicine, Taipei Medical University, Taipei 11031, Taiwan; yuchengku@tmu.edu.tw; 4School of Post-Baccalaureate Chinese Medicine, College of Chinese Medicine, China Medical University, Taichung 40402, Taiwan; 5Department of Pharmacy Practice, Tri-Service General Hospital, School of Pharmacy, National Defense Medical Center, Taipei 11490, Taiwan; tangling@mail.ndmctsgh.edu.tw; 6Department of Rheumatology/Immunology and Allergy, Department of Medicine, Tri-Service General Hospital, National Defense Medical Center, Neihu Dist., Taipei City 114, Taiwan; lfc10399@mail.ndmctsgh.edu.tw; 7School of Pharmacy, National Defense Medical Center, Taipei 11490, Taiwan; shiang-jiun@803.org.tw; 8Graduate Institute of Medical Sciences, National Defense Medical Center, Taipei 11490, Taiwan; 9The PhD Program of Translational Medicine, College of Medical Science and Technology, Taipei Medical University, Taipei 11031, Taiwan; 10Clinical Research Center, Taipei Medical University Hospital, Taipei Medical University, Taipei 11031, Taiwan; 11TMU Research Center of Cancer Translational Medicine, Taipei Medical University, Taipei 11031, Taiwan; 12PhD Program in Drug Discovery and Development Industry, College of Pharmacy, Taipei Medical University, Taipei 11031, Taiwan

**Keywords:** glioblastoma multiforme (GBM), temozolomide (TMZ), chemoradioresistance, genetic heterogeneity, bioinformatics, molecular docking, National Cancer Institute (NCI)-60

## Abstract

**Simple Summary:**

Glioblastoma multiforme (GBM) remains to be the most frequent malignant tumor of the central nervous system (CNS), which accounts for approximately 54% of all primary brain gliomas. Current treatment modalities for GBM include surgical resection, followed by radiotherapy and chemotherapy with temozolomide (TMZ). However, due to its genetic heterogeneity, GBM tumors always recur, due to treatment reasistance. The aim of this study was to identify molecular gene signatures, responsible for cancer initiation, progression, resistances and to treatment, metastasis, and also evaluate the potency of our novel compounds SJ10 as potential target for *CCNB1/CDC42/MAPK7/CD44* oncogenic signatures. Accordingly, we used computational simulation and identify these signatures as regulators of the cell cycle in GBM, which leads to cancer development and metastasis. We also showed the antiproliferative and cytotoxic effects of SJ10 compound against a panel of NCI-60 cancer cell lines. This suggests the potential of the compounds to inhibit *CCNB1/CDC42/MAPK7/CD44* in GBM.

**Abstract:**

Current anticancer treatments are inefficient against glioblastoma multiforme (GBM), which remains one of the most aggressive and lethal cancers. Evidence has shown the presence of glioblastoma stem cells (GSCs), which are chemoradioresistant and associated with high invasive capabilities in normal brain tissues. Moreover, accumulating studies have indicated that radiotherapy contributes to abnormalities in cell cycle checkpoints, including the G_1_/S and S phases, which may potentially lead to resistance to radiation. Through computational simulations using bioinformatics, we identified several GBM oncogenes that are involved in regulating the cell cycle. Cyclin B1 (*CCNB1*) is one of the cell cycle-related genes that was found to be upregulated in GBM. Overexpression of *CCNB1* was demonstrated to be associated with higher grades, proliferation, and metastasis of GBM. Additionally, increased expression levels of CCNB1 were reported to regulate activation of mitogen-activated protein kinase 7 (*MAPK7*) in the G_2_/M phase, which consequently modulates mitosis; additionally, in clinical settings, *MAPK7* was demonstrated to promote resistance to temozolomide (TMZ) and poor patient survival. Therefore, *MAPK7* is a potential novel drug target due to its dysregulation and association with TMZ resistance in GBM. Herein, we identified *MAPK7*/extracellular regulated kinase 5 (*ERK5*) genes as being overexpressed in GBM tumors compared to normal tissues. Moreover, our analysis revealed increased levels of the cell division control protein homolog (*CDC42*), a protein which is also involved in regulating the cell cycle through the G_1_ phase in GBM tissues. This therefore suggests crosstalk among *CCNB1/CDC42/MAPK7/*cluster of differentiation 44 (*CD44*) oncogenic signatures in GBM through the cell cycle. We further evaluated a newly synthesized small molecule, SJ10, as a potential target agent of the *CCNB1/CDC42/MAPK7/CD44* genes through target prediction tools and found that SJ10 was indeed a target compound for the above-mentioned genes; in addition, it displayed inhibitory activities against these oncogenes as observed from molecular docking analysis.

## 1. Introduction

Glioblastoma multiforme (GBM) is one of the most frequent malignant tumors of the central nervous system (CNS) [1,2], which accounts for approximately 54% of all primary brain gliomas, with a yearly incidence of 3.2 per 100,000 adults globally [3,4], and is classified as grade IV by the World Health Organization (WHO) [5]. It is associated with poor clinical outcomes, with fewer than 10% of patients reaching a 5-year survival rate after diagnosis [6,7]. The current treatment modalities for GBM include surgical resection, followed by radiotherapy and chemotherapy with temozolomide (TMZ) [8,9,10]. However, due to genetic heterogeneity, GBM tumors always recur mainly at the resection site, leading to an overall median survival of only 15 months following the initial diagnosis [5,11]. Therefore, understanding molecular mechanisms and invasive characteristics of GBM is pivotal as an essential strategy for developing more-effective therapeutics. Resistance to treatment in GBM is also associated with glioblastoma stem cells (GSCs), which may potentially assist GBM cancer cells to escape irradiation [11,12,13]. Studies showed that GCSs are resistant to TMZ chemotherapy, thus promoting radio resistance through DNA damage response activation [11].

One of the stem cell markers that is commonly expressed in various cancer types, including GBM is cluster of differentiation 44 (CD44), a surface adhesion receptor which promotes cancer progression and metastasis [14]; its expression in GBM cells is also crucial for GBM invasion and migration [15]. *CD44*-expressing cells were shown to escape exogenous DNA damage from radiation-induced double-stranded breaks (DSBs); it ultimately promoted tumor recurrence and resistance to radiation [16]. Moreover, accumulating studies indicated that radiotherapy contributes to abnormalities in cell cycle checkpoints, including the G_1_/S and S phases, which may potentially lead to resistance to radiation [17,18]. Cyclin B1 (*CCNB1*) is one of the cell cycle-related genes that was reported to be a potential biomarker in GBM [19,20]. Overexpression of *CCNB1* was demonstrated to be associated with higher grades, proliferation, and metastasis of GBM [21]. Additionally, *CCNB1* was shown to regulate cell mitosis at the G_2_/M phase through interacting with cyclin-dependent kinase 1 (*CDK1*) [22]. Thus, *CCNB1/CDK1* may be potential diagnostic and prognostic markers of GBM [23,24,25].

Mitogen-activated protein kinase 7 (*MAPK7*) is a member of *MAPK*s, which regulate signaling transduction cascades [26] and are associated with multiple cellular processes, such as cell proliferation and survival [27,28]. High expression levels of *MAPK7* were identified in GMB tumors compared to adjacent normal brain tissues. In clinical settings, *MAPK7* promotes resistance to TMZ and poor patient survival; however, its role in GBM still remains to be further investigated [29]. Increased expression levels of CCNB1 were reported to regulate activation of *MAPK7* in the G_2_/M phase, which consequently modulates mitosis [28]. Therefore, *MAPK7* is a potential novel drug target due to its dysregulation and association with TMZ resistance in GBM [30,31,32]. Moreover, cell division control protein 42 homolog (*CDC42*) is a protein which is also involved in regulating the cell cycle through the G_1_ phase. Overexpression of *CDC42* in GBM was demonstrated to promote tumor cell invasion and migration; additionally, *CDC42* was associated with low survival rates and drug resistance in GBM patients [33,34]. This therefore suggests crosstalk among *CCNB1/CDC42/MAPK7/CD44* oncogenic signatures in GBM through the cell cycle.

Integrated bioinformatics analyses have been extensively applied in the early stages of drug discovery and development and have significantly accelerated the process, as well as reduced costs. Computational simulation approaches and molecular structural analyses of ligand-protein interactions have contributed to the identification and prediction of novel diagnostic and prognostic biomarkers in cancer research [35,36]. In this study, we explored Microarray Data Extraction and predicted overexpressed and downregulated genes in GBM tumors; moreover, we utilized online prediction tools to further identify and validate expressions of our genes of interest, the *CCNB1/MAPK7/CDC42/CD44* oncogenes and also predicted patients’ clinical outcomes in GBM under the same settings. To date, only bevacizumab, everolimus and TMZ are the common FDA-approved drugs for brain tumor treatment [37]. Fortunately, with our innovative lab techniques, as mentioned in previous preliminary studies in drug discovery [38,39], we synthesized 9-chloro-6-(piperazin-1-yl)-11H-indeno[1,2-c]quinolin-11-one (SJ10), a quinolone and piperazine derivative, with anticancer activities (Figure 1). In addition to our earlier studies, quinoline derivatives were demonstrated to possess anticancer activities by inducing DNA double-strand breaks and apoptosis [39,40]. Therefore, in this study, we performed drug target predictions and identified *CCNB1/MAPK7/CDC42/CD44* oncogenic signatures as potential drug candidates of SJ10, and further performed ligand-protein binding simulations using in silico molecular docking, which validated *CCNB1/MAPK7/CDC42/CD44* as druggable candidates of SJ10. The antiproliferative and cytotoxic effects of SJ10 were evaluated in vitro using the US National Cancer Institute (NCI)-60 central nervous system (CNS) cell lines to determine responses of single-dose and dose-dependent treatments with SJ10 [41].

## 2. Material and Methods

### 2.1. Dataset Collection

Gene expression profiles (GEPs) from GBM patient samples were downloaded from the Gene Expression Omnibus (GEO) database (http://www.ncbi.nlm.nih.gov/gds/, accessed on 4 August 2021), an international public repository for high-throughput microarray data [42]. (3) different expression profile datasets obtained, viz., GSE4290, GSE68848 and GSE30563 were further analyzed using GEO2R (https://www.ncbi.nlm.nih.gov/geo/geo2r/, accessed on 4 August 2021), an interactive online platform to identify differentially expressed genes (DEGs) [43], which was used to identify DEGs between GBM tumor samples and normal samples. The Benjamini–Hochberg adjustment was made to *p* values (adj. *p*) to control the false discovery rate (FDR) and maintain the balance between the possibility of false-positives and the detection of significant genes. The fold-change (FC) threshold was set to 1.5, and adj. *p* < 0.05 was considered statistically significant. Venn diagrams were constructed using the Bioinformatics & Evolutionary Genomics (BEG) online tool (http://bioinformatics.psb.ugent.be/webtools/Venn/, accessed on 4 August 2021).

### 2.2. Identifying Molecular Targets and Therapeutic Classes of SJ10

Potential SJ10 target genes were predicted using an open-source web tool based on the Prediction of Biological Activity Spectra (PASS) [44] (http://www.way2drug.com/passonline/predict.php, accessed on 17 August 2021). In addition, we explored the Swiss target prediction tool (http://www.swisstargetprediction.ch/, accessed on 17 August 2021), a web-based algorithm that uses the principle of similarity to predict drug targets of bioactive small molecules [45,46] as an independent tool to further validate the predicted potential target genes of SJ10 (Table 1).

### 2.3. DEG Identification by the Tumor Immune Estimation Resource (TIMER)

Expression profiles of genes showing differential expression between GBM tumor and adjacent non-tumor tissues in The Cancer Genome Atlas (TCGA) database were analyzed with TIMER (https://cistrome.shinyapps.io/timer/, accessed on 9 August 2021), a web-based tool for the analysis of interactions between genes of interest and immune cells. The relative gene expression level is indicated as transcripts per million (TPM) and the expression value was normalized by log transformation. Moreover, we explored the Chinese Glioma Genome Atlas (CGGA) (http://www.cgga.org.cn/index.jsp, accessed on 16 August 2021) to analyze gene expression correlations between the two datasets into positive and negative correlations, with positive Pearson correlation coefficients and *p* < 0.05 considered statistically significant.

### 2.4. Validation of DEGs in GBM

To validate expression levels of identified DEGs in GBM, we explored the Human Protein Atlas (HPA) database for immunohistochemistry (IHC) (https://www.proteinatlas.org/, accessed on 13 September 2021) to compare expression levels between tumor samples and normal samples. The HPA database represents the protein expression in 44 major human tissues and some cancer tissues by IHC [47]. Statistical analyses were performed using the statistical package for social sciences (SPSS) vers. 21.0 (Chicago, IL, USA) and the *p* value was determined using the Mann–Whitney U-test. Moreover, for further analysis, the predicted genes were validated by an independent bioinformatics tool, the CGGA [48].

### 2.5. Protein-Protein Interaction (PPI) Network Construction and Functional Enrichment Analysis

To assess PPIs, we used the search tool for the Retrieval of Interacting Genes/Proteins database (STRING, https://string-db.org/, accessed on 21 September 2021), a web tool developed to analyze interactions of PPIs, such as physical and functional associations [49]. Functional enrichments with the clustering network were also retrieved from the STRING analysis, and they included gene ontology (GO) involving biological processes (BPs) and Kyoto Encyclopedia of Genes and Genomes (KEGG) pathways, with *p* < 0.05 considered significant. For further analysis, we used Network Analyst (https://www.networkanalyst.ca/, accessed on 21 September 2021), a web-based visual analytics platform for comprehensive gene and protein expression profiling [50,51]. In this platform, we used the SIGnaling Network Open Resource (SIGNOR 2.0) and selected the BP database to analyze enriched co-expressed genes.

### 2.6. Predictions of Patient Clinical Outcomes with Radiomics Signature Construction

To predict prognostic outcomes in GBM, radiomics signatures were constructed using the GlioVis database (http://gliovis.bioinfo.cnio.es/, accessed on 28 August 2021), an online portal used for analysis of brain tumor expression [52]. The distribution of radscores and maximally selected rank statistics of DEGs were used to determine the optimum cutoff values for the *CCNB1, CDC42, MAPK7,* and *CD44* oncogenes, in order to evaluate overall survival (OS).

### 2.7. Receiver Operating Characteristic (ROC) Curves and Kaplan-Meier (KM) Analyses Were Used to Validate the Prognostic Values of the CCNB1, CDC42, MAPK7, and CD44 Oncogenic Signatures in GBM Samples

To evaluate and validate the diagnostic and prognostic significance of *CCNB1, CDC42, MAPK7,* and *CD44* in GBM patients, we used ROC curve, which was retrieved from (https://kmplot.com/analysis/, accessed on 1 September 2021), and further explored the GlioVis database for the KM analysis. The ROC curve was based on true positive (sensitivity) and false positive (specificity) rates in GBM patients. We evaluated whether the test measurement had a specific condition. We assessed the area under the curve (AUC), and an AUC of 0.5 indicated no discrimination, while an AUC of 1.0 indicated discrimination of the curve that includes all possible decision thresholds from a diagnostic test result, which were patients who experienced disease onset and individuals who did not.

### 2.8. Evaluation of Drug Likeness, Pharmacokinetics (PKs), and Medicinal Chemistry of SJ10

Identifying novel and potential drug candidates in the early stage of drug discovery and development is crucial, as it reduces time and costs; herein, we applied the drug-likeness concept based on specific criteria [53,54]. We explored the SwissADME algorithm developed by the Swiss Institute of Bioinformatics (http://www.swissadme.ch/index.php, accessed on 2 September 2021), and molecular in silico (molsoft) tools (https://molsoft.com/mprop/, accessed on 2 September 2021), to evaluate the PKs, drug likeness, medicinal chemistry friendliness, adsorption, distribution, metabolism, excretion, and toxicity (ADMET) properties of SJ10 [55,56]. We analyzed the drug-likeness properties according to the Lipinski (Pfizer) rule-of-five), Ghose (Amgen), Veber (GSK), and Egan (Pharmacia), and further showed relationships between PK and physicochemical properties [57]. Moreover, we analyzed the gastrointestinal absorption (GIA) and brain-penetration properties using the brain or intestinal estimated permeation (BOILED-Egg) model [58]. The Abbot Bioavailability Score was determined based on the probability of the compound having at least 10% oral bioavailability in rats or measurable Caco-2 permeability [59].

### 2.9. In Vitro Anticancer Screening of SJ10 against NC1-60 CNS Cells

SJ10 was submitted to the National Cancer Institute (NCI)-Development Therapeutics Program (DTP) to be screened for potential antiproliferative and cytotoxic effects against a panel of NCI-60 CNS cell lines, in agreement with the outlined protocol of the NCI (https://dtp.cancer.gov/, accessed on 11 September 2021). The compound was tested at an initial dose of 10 μM. Results showed that SJ10 exhibited antiproliferative activities against CNS cell lines.

### 2.10. Molecular Docking Analysis

Receptor-ligand interactions were predicted using a molecular docking analysis, a technique used to predict the predominant binding ability of a ligand with a protein’s three-dimensional (3D) structure [60]. To assess possible interactions of SJ10 with target genes predicted and selected from the Swiss-target and PASS prediction tools, we performed a docking analysis of SJ10 with the *CCNB1, CDC42, MAPK7*, and *CD44* oncogenes. For further analysis, we used the Food and Drug Administration (FDA)-approved standard inhibitors of *CDC42* and *MAPK* of CASIN and BAY-885, respectively. Accordingly, the 3D structure of SJ10 was assembled with the Avogadro molecular visualization tool [61], the 3D structures of CASIN (CID:2882155) and BAY-885 (CID:134128280) were downloaded from PubChem as SDF files, and the files were subsequently converted to PDB format using PyMol software (https://pymol.org/2/, accessed on 6 October 2021). In addition, the crystal structures of CCNB1 (PDB:2B9R), CDC42 (PDB: 2ODB), MAPK7 (PDB:4H3Q), and CD44 (PDB:1UUH) were downloaded from the Protein Data Bank (PDB). For further processing, we converted all PDB files to PDBQT file format using autodock software (http://autodock.scripps.edu/resources/adt, accessed on 6 October 2021) and, finally, performed docking. To visualize and interpret the docking results, we applied BIOVIA discovery studio software for analysis [62].

### 2.11. Statistical Analysis

Pearson’s correlations were used to assess correlations of *CCNB1/CDC42/MAPK7/CD44* expressions in GBM cancer types. The statistical significance of DEGs was evaluated using the Wilcoxon test. * *p* < 0.05 was accepted as being statistically significant.

## 3. Results

### 3.1. Identification of DEGs in GBM

Gene expression profiles (GEPs) from GBM samples and normal brain samples tallied from different studies were extracted from the microarray dataset. The analytical results showed that 100, 256, and 30 GBM samples and normal samples were, respectively, obtained from the GSE4290, GSE68846, and GSE30563 datasets. Further analysis with a Venn diagram demonstrated 87 overlapping upregulated genes from the three datasets (Figure 2A) and 50 overlapping downregulated genes from the same database (Figure 2B). Moreover, Figure 1 is the heatmap of overexpressed overlapping genes. Figure 2D–F shows volcano plots of GBM tumor samples compared to normal samples, the volcano plot revealed the statistical significance of the difference between tumor and normal samples through −10 log and −2 log fold change, respectively. The *p*-value in the volcano plot was used to indicate threshold indicators for adjusted *p*-values, which was further used to show all the genes that are statistically differentially-expressed with an adjusted *p* value threshold of 0.05 considered significant.

### 3.2. Evaluation of Drug Likeness, PKs, and Medicinal Chemistry of the SJ10 Compou

We explored the SwissADME algorithm developed by the Swiss Institute of Bioinformatics and molecule in silico (molsoft) to evaluate the PKs, drug likeness, medicinal chemistry friendliness, adsorption, distribution, metabolism, excretion, and toxicity (ADMET) properties of SJ10 [55,56]. We analyzed the drug-likeness properties according to the Lipinski (Pfizer) rule-of-five)), Ghose (Amgen), Veber (GSK), and Egan (Pharmacia), and further showed relationships between the PK and physicochemical properties [57]. Moreover, we analyzed the GIA and brain-penetration properties, using the brain or intestinal estimated permeation (BOILED-Egg) model [58]. The Abbot Bioavailability Score was determined based on the probability of the compound having at least 10% oral bioavailability in rats or measurable Caco-2 permeability [59] (Figure 3).

### 3.3. CCNB1/CDC42/MAPK7/CD44 Oncogenic Signatures Are Overexpressed in GBM

A bioinformatics analysis through TIMER online web tool with default settings showed significantly increased messenger (m)RNA levels of *CCNB1/CDC42/MAPK7/*CD44 in pan cancers, including GBM tumor tissues compared to normal tissues from TCGA) (Figure 4A–D). Relative gene expression levels are indicated as transcripts per million (TPM) and the expression value was normalized by log transformation of the statistical significance, as evaluated by the Wilcoxon test, with *p* value significant codes: 0 ≤ ******* < 0.001 ≤ ****** < 0.01 ≤ ***** < 0.05 ≤ . < 0.1. We further explored the CGGA tool with default settings to investigate correlations among the *CCNB1/CDC42/MAPK7/CD44* oncogenes. When all four genes were combined for analysis, the predicted results showed positive correlations ranging *r* = 0.43~0.67 of CCNB1 with CDC42, CCNB1 with MAPK7, CCNB1 with CD44, and MAPK7 with CD44 in GBM patients (Figure 4E–H), with positive Pearson correlation coefficients and *p* < 0.05 considered statistically significant.

### 3.4. Validation of CCNB1/CDC42/MAPK7/CD44 Oncogenic Signature Expressions in GBM

To validate expression levels of the *CCNB1/CDC42/MAPK7/CD44* gene signatures in GBM, we explored the HPA database for IHC to compare gene expression levels between GBM tumor tissues and normal samples. CCNB1 displayed medium staining, with strong intensity and quantity (25%) (Figure 5A), while CDC42 displayed medium staining, with moderate intensity and quantity (75%) (Figure 5B), and MAPK7 and CD44 displayed high staining with strong intensity and quantity (75%) (Figure 5C,D) in GBM tissues as compared to normal tissues. For further analysis, the GlioVis database showed increased mRNA expression levels of *CCNB1/CDC42/MAPK7/CD44* oncogenes in GBM tissues compared to non-tumor tissues (Figure 5E–H). In addition, we explored the CGGA, an independent glioma database, and validated expressions of the *CCNB1/CDC42/MAPK7/CD44* gene signatures in WHO grade II, III, and IV GBM tumors using the Analysis of variance (ANOVA) (Figure 5I–L), with *p* < 0.05 considered statistically significant.

### 3.5. Immunofluorescent (IF) Staining of the U251-MG GBM Human Cell Line

To further validate expressions of the *CCNB1/CDC42/MAPK7/CD44* genes in GBM, we explored HPA IF staining, using the U251-MG GBM cell line. The following antibodies were used for staining: *CCNB1* (HPA030741), *CDC42* (CAB004360), *MAPK7* (CAB018561), and *CD44* (CAB000112). Staining results of the U251-MG cell line exhibited the location of genes, with antibodies shown in green, nuclei in blue, and microtubules in red. CCNB1 was localized in the cytosol, CDC42 was detected in microtubules, while the localization of MAPK7 was in the nucleoplasm and CD44 was found in plasma membranes (Figure 6).

### 3.6. PPI Network Construction and Functional Enrichment Analysis

To assess PPIs, we used the STRING database (https://string-db.org/, accessed on 21 September 2021), a web tool developed to analyze interactions of PPIs, such as physical and functional associations. The clustering analysis had nine nodes and 15 edges, with an average local clustering coefficient of 0.917 and a PPI enrichment *p* value of 0.0293. Moreover, the interaction score confidence was set to > 0.4, and considered most significant. Active interactions were based on text mining, experiments, databases, co-expressions, neighborhood, gene fusion and co-occurrence (Figure 7A). Functional enrichments with the clustering network were also retrieved from the STRING analysis, and they included gene ontology (GO) involving BPs and KEGG pathways, with *p* < 0.05 considered significant (Figure 7B,C). For further analysis, we used Network Analyst (https://www.networkanalyst.ca/, accessed on 21 September 2021), a web-based visual analytics platform for comprehensive gene and protein expression profiling [50,51]. In this platform, we used the SIGnaling Network Open Resource (SIGNOR 2.0) and selected the BP database to analyze enriched co-expressed genes (Figure 7D). The signaling network analysis of KEGG pathway enrichment showed co-expressions of CCNB*1/CDC42/MAPK7/CD44* oncogenes in the same network cluster, and results were viewed from the network topology in a force atlas layout analyzed from the Igraph R package (Figure 7E).

### 3.7. Predictions of Patient Clinical Outcomes with Radiomics Signature Construction

Prognostic outcomes of GBM patients were predicted by exploring radiomics signatures constructed using the GlioVis database, and distributions of Radscores and maximally selected rank statistics were used to determine optimal cutoff values for the *CCNB1, CDC42, MAPK7,* and *CD44* oncogenes. The obtained cutoff scores (Radscores) were 2.83, 6.62, 3.9, and 3.48, respectively (Figure 8A–H). This analysis therefore showed that patients with lower Radscores generally displayed better OS; however, since the *CCNB1/CDC42/MAPK7/CD44* oncogenes were shown to be highly expressed in GBM, herein, they also exhibited high Radscores and, consequently, worse prognoses. Therefore, predicted expressions of the *CCNB1, CDC42, MAPK7* and *CD44* oncogenes exhibited significant roles in the cell cycle, and thus are potential prognostic biomarkers in GBM.

### 3.8. High Expressions of CCNB1, CDC42, MAPK7, and CD44 Were Associated with a Poor Prognosis in GBM

To evaluate and validate prognostic significant values of *CCNB1, CDC42, MAPK7*, and *CD44* in GBM patients, we used an ROC curve and KM analysis. The ROC curve was based on true (sensitive) and false (selective) positive rates of responses in GBM patients. AUC scores of *CCNB1, CDC42, MAPK7*, and *CD44* were 0.534, 0.515, 0.541, and 0.538, respectively (Figure 9A–D). The KM analysis and log-rank test showed significantly prolonged OS times in the low-risk group compared to the high-risk group, with each subtype displaying different cutoff values; *p* < 0.05 was considered statistically significant (Figure 9E–H). This indicated that the *CCNB1, CDC42, MAPK7,* and *CD44* oncogenic signatures possessed potential diagnostic abilities in GBM. To evaluate whether the test measurements had specific conditions, we assessed the AUC, and an AUC of 0.5 indicated no discrimination, while an AUC of 1.0 indicated discrimination. The curve that included all possible decision thresholds from a diagnostic test result were patients who had experienced disease onset and individuals who had no thresholds from a diagnostic test result, which were patients who had experienced disease onset and individual thresholds.

### 3.9. In Vitro Anticancer Screening of SJ10 against NC1-60 CNS Cell Lines

SJ10 was submitted to the NCI-DTP for screening for potential antiproliferative and cytotoxic effects against a panel of NCI-60 CNS cell lines, in agreement with the outlined protocol of the NCI. The compound was tested at an initial dose of 10 μM. Results showed that SJ10 exhibited antiproliferative activities against several CNS cell lines. The compound growth inhibition (GI) percentage showed that SNB-19 cells were more sensitive, with GI of 67.25%, followed by SF-539 at 36.91%, SF-268 at 34.33%, SF-295 at 23.75%, and SNB-75 at 22.34%, as shown in Figure 10A. The compound was further evaluated with dose-dependent treatment, since it exhibited antiproliferative activities at an initial dose of 10 μM. Accordingly, SF-268 displayed complete growth inhibition (−100%), followed by U251 at −96%, SNB-75 at −84%, SF-539 at −79%, SF-295 at −76%, and SNB-19 at −42%. Sulforhodamine B (SRB) dual-pass staining was used to further investigate the in vitro 50% growth inhibition (GI_50_)/50% inhibitory concentration (IC_50_), and results ranged 1.14~2.15 μM in the CNS cell lines, with SNB-75 more sensitiv at 1.14 μM, followed by U251 at 1.59 μM, SF-268 at 1.64 μM, SNB-19 at 1.67 μM, SF-539 at 1.69 μM, and SF-295 at2.15 μM, showing a smaller response to SJ10 (Figure 10B,C).

### 3.10. Molecular Docking Analysis

The potential inhibitory effects of SJ10 were assessed using a docking analysis. Results of receptor-ligand interactions obtained from the Autodock tool revealed putative binding affinities of SJ10 with CCNB1 (−7.9 kcal/mol), CDC42 (−7.8 kcal/mol), MAPK7 (−8.4 kcal/mol), and CD44 (−7.0 kcal/mol). When compared to standard inhibitors of CASIN (CID: 2882155) for CDC42 and BAY-885 (CID: 134128280) for MAPK7, they showed lower binding energies of −7.4 and −7.3 kcal/mol, respectively. For a further analysis, we used Pymol and Discovery Studio to visualize the analytical results. The SJ10/CCNB1 complex displayed interactions by conventional hydrogen (H) bonds with ALA128 (2.07 Å) and ARG68 (2.73 Å). The interactions were stabilized by van der Waals interactions (ASN130, LEU129, PHE131, GLY132, PHE131, ASN130, GLY134, and PRO136), pi-sigma (GLY132), and pi-alkyl (LEU17, LEU17, and ARG135) displayed in their binding pockets. The SJ10/CDC42 complex displayed van der Waals interactions (THR25, PHE28, SER30, THR17, and TYR40), pi-sigma (ILE21), and pi-alkyl (PHE18, LYS27, and PRO29) in their binding pockets, while SJ10/MAPK7 exhibited conventional hydrogen bond with SER153 (2.23 Å) and was further stabilized by van der Waals interactions (LYS114, GLY34, TRP192, and THR193), carbon hydrogen bond (GLU33), pi-sigma (THR190), pi-pi stacked (TYR113), and pi-alkyl (PRO152 and LYS151) in their binding pockets. Interactions between the SJ10/CD44 complex displayed van der Waals interactions (THR102, GLY103, ARG90, LEU70, TYR79, SER71, ILE96, and ARG78), carbon hydrogen bond (CYS77), pi-pi T-shaped (TYR42) and pi-alkyl (ILE91) in their binding pockets (Figure 11). For further analysis, we used the FDA approved standard inhibitors of *CDC42* and *MAPK7*, CASIN and BAY-885 respectively. The interaction of CDC42 in complex with CASIN exhibited binding energy of (−7.4 kcal/mol) and MAPK7 in complex with BAY-885 displayed binding energy of (−7.3 kcal/mol), these results exhibited a much lower binding affinities as compared to SJ10, this suggesting the potential inhibitory effects of SJ10 in GBM expression CCNB1, CDC42, MAPK7, and CD44 oncogenic signatures (Figure 12, Table 2).

## 4. Discussion

Despite improvements in standard therapies, including surgical resection, radiation, and chemotherapy with TMZ, patients with GBM still exhibit poor clinical outcomes, with a median survival of only about 15 months [63], mainly due to GBM’s biological and genetic heterogeneity. Therefore, understanding molecular mechanisms and invasive characteristics of GBM is pivotal as an essential strategy for developing more-effective therapeutics. Integrated bioinformatics analyses have been extensively applied in the early stages of drug discovery and development and have significantly accelerated the process and reduced costs. In the current study, we applied computational simulation analyses to predict and identify dysregulated gene networks and pathways leading to resistance to radio- and chemotherapies (TMZ). Accumulating studies have demonstrated that therapeutic resistance in GBM is also associated with GSCs, which may potentially assist GBM cancer cells escape irradiation. Others have shown that GCSs are resistant to TMZ chemotherapy, thus promoting radioresistance through DNA damage-response activation [11,12,13].

SJ10 (NSC7772862) is a small molecule and a derivative of a quinolone and piperazine derivative and was recently synthesized in our laboratory. Through application of the Swisstarget and PASS prediction tools (Table 1), we predicted *CCNB1, CDC42, MAPK7*, and *CD44* oncogenic signatures as target genes for SJ10. Moreover, we explored the SwissAMDE and molsoft algorithms to evaluate the PK, drug-likeness, medicinal chemical friendliness, and ADMET properties of SJ10 [55,56]. The compound successfully passed the required physicochemical properties, medicinal chemistry, PK, and drug-likeness criteria. Bioavailability radar, displaying the six physicochemical properties of absorption-included lipophilicity (XLOGP3 = 3.90), molecular weight (349.10 g/mol), polarity (PSA = 37.08 Å^2^), solubility (Log S (ESOL) = −4.7), flexibility (rotation = 4), saturation (Fraction Csp3 = 0.2), and pKa (=0.5) of the SJ10 compound. In addition, the SJ10 compound demonstrated highly probable GIA absorption, a bioavailability score (55%), and good synthetic accessibility (2.89). The compound reached the BBB with a score of 4.98, and further displayed a drug-like model score of (−0.68) (Figure 2).

We identified significantly increased mRNA levels of the *CCNB1/CDC42/MAPK7/CD44* oncogenes in pan cancers, including GBM tumor tissues compared to normal tissues from TCGA, using the TIMER bioinformatics tool. These results were further validated using the HPA and GlioVis database analyses, which showed similar outputs displaying overexpression of *CCNB1/CDC42/MAPK7/CD44* gene signatures in WHO grade II, III, and IV GBM tumors using an ANOVA. For further analysis, we used the STRING online web tool and showed that *CCNB1/CDC42/MAPK7/CD44* actively interacted with each other in the same clustering network, based on text mining, experiments, databases, co-expressions, neighborhood, gene fusion, and co-occurrence and also exhibited enrichment of GO involving BPs and (KEGG pathways, with *p* < 0.05 considered significant (Figure 6B,C). Furthermore, we predicted patients’ clinical outcomes using the Radiomics signature constructed from the GlioVis database, to determine optimal cutoff values for the *CCNB1, CDC42, MAPK7,* and *CD44* oncogenes. The obtained cutoff scores (Radscore) were 2.83, 6.62, 3.9, and 3.48, respectively (Figure 7). The analysis therefore showed that patients with lower Radscores generally displayed better OS; however, since the *CCNB1/CDC42/MAPK7/CD44* oncogenes were shown to be highly expressed in GBM, herein, they also exhibited high Radscores, which consequently led to worse prognoses. Therefore, predicted expressions of the *CCNB1, CDC42, MAPK7,* and *CD44* oncogenes exhibited significant roles in the cell cycle, and thus are potential prognostic biomarkers for GBM. In addition, *MAPK7* is a potential novel drug target due to its dysregulation and association with TMZ resistance in GBM. Herein, we showed that targeting *MAPK7* in GBM tumors can potentially improve the strength of TMZ in suppressing tumor cells [30,31,32].

The potential anticancer activities of SJ10 were evaluated against NCI human CNS cell lines. Accordingly, an initial dose of 10 μM exhibited antiproliferative activities against the CNS cell lines, as shown in Figure 9A. The compound was further evaluated with dose-dependent treatment, since it exhibited antiproliferative activities at an initial dose of 10 μM. Accordingly, SJ10 displayed complete growth inhibition at −100% against SF-268 cells, followed by U251 at −96%, SNB-75 at −84%, SF-539 at −79%, SF-295 at −76%, and SNB-19 at −42%. SRB dual-pass staining was used to further investigate in vitro GI_50_/IC_50_ values, and results ranged 1.14~2.15 μM in the CNS cell lines, with SNB-75 more sensitive at 1.14 μM, followed by U251 at 1.59 μM, SF-268 at 1.64 μM, SNB-19 at 1.67 μM, SF-539 at 1.69 μM, and SF-295 at 2.15 μM, showing a weaker response to SJ10. Finally, we evaluated the potential inhibitory effects of SJ10, which were assessed using a docking analysis. The results of receptor-ligand interactions obtained from the Autodock tool revealed higher binding energies of SJ10 with CCNB1 (−7.9 kcal/mol), CDC42 (−7.8 kcal/mol), MAPK7 (−8.4 kcal/mol), and CD44 (−7.0 kcal/mol) compared to the standard inhibitors of CASIN (CID: 2882155) for CDC42 and BAY-885 (CID: 134128280) for MAPK7, which showed lower respective binding energies of −7.4 and −7.3 kcal/mol (Fig.12). Therefore, the above-mentioned results suggest that SJ10 exhibits drug-like characteristics, with anticancer activities and is a potential oral drug candidate. Further in vitro and in vivo studies are both currently in progress in our laboratory.

## 5. Conclusions

In summary, our obtained results showed that *CCNB1/CDC42/MAPK7/CD44* oncogenic signatures are potential biomarkers of GBM therapeutic-resistant tumors, and potential drug targets of our novel small molecule, SJ10. We further showed that SJ10 exhibits anticancer activities against a panel of NCI human CNS cancer cell lines when administered at an initial dose of 10 μM and also in a dose-dependent manner. We evaluated receptor-ligand interactions using a docking analysis and identified unique and higher binding energies of SJ10 in complex with the *CCNB1, CDC42, MAPK7,* and *CD44* oncogenes, compared to their interactions with two FDA-approved inhibitors. Further in vitro and in vivo studies are both currently in progress in our laboratory.

## Figures and Tables

**Figure 1 cancers-14-00262-f001:**
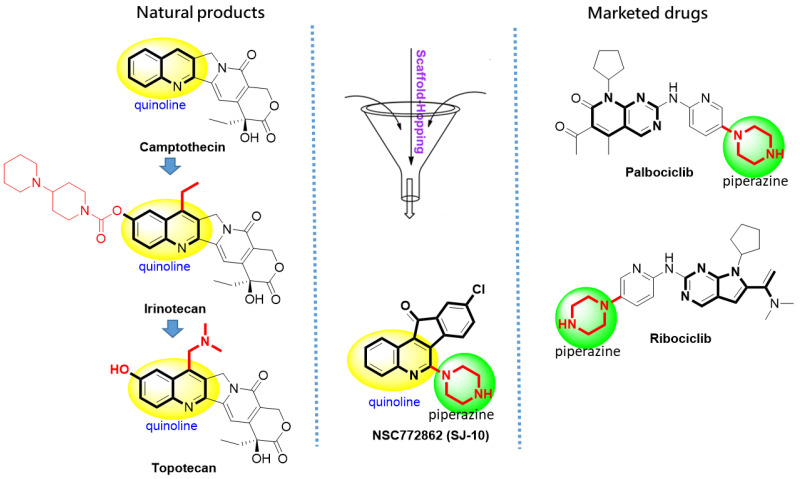
Rationale of NSC772862 (SJ10) and some of the representative drugs in natural products and marketed drugs.

**Figure 2 cancers-14-00262-f002:**
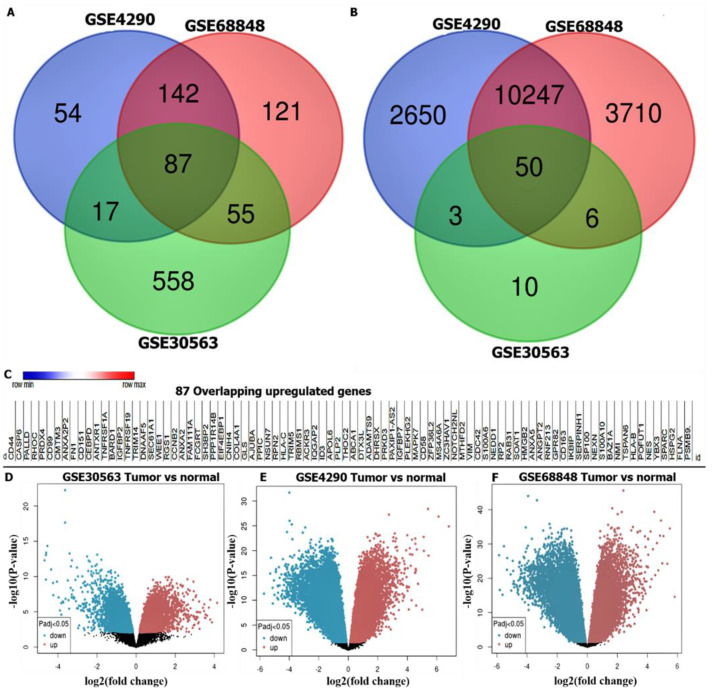
Differentially-expressed genes (DEGs) in glioblastoma multiforme (GBM) extracted from the GSE4290, GSE68846, and GSE30563 microarray datasets. (**A**) Venn diagram of 87 selected overexpressed overlapping DEGs. (**B**) Venn diagram of 50 selected downregulated overlapping DEGs. (**C**) is the heatmap of overexpressed overlapping genes. (**D**–**F**) Volcano plots of DEGs from the GSE4290, GSE68848 and GSE30563 datasets with red and blue dots respectively representing upregulated and downregulated genes (at *p* < 0.05).

**Figure 3 cancers-14-00262-f003:**
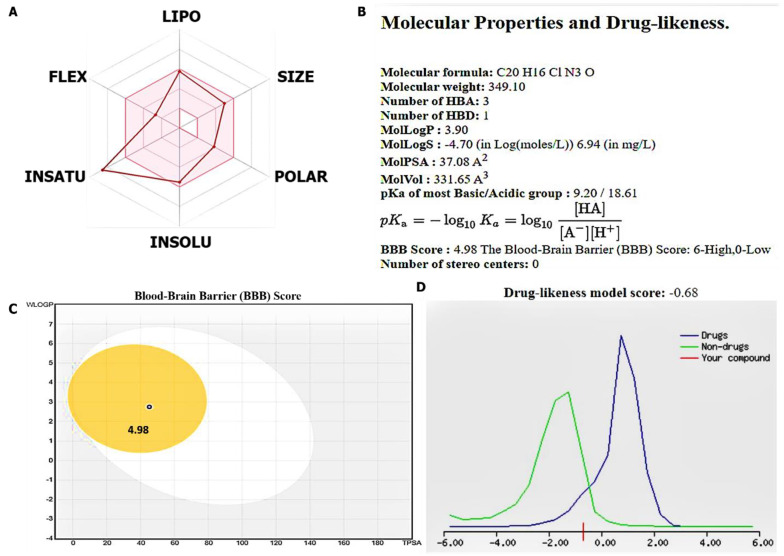
SJ10 passed the required physicochemical properties, medicinal chemistry, pharmacokinetics (PK), and drug-likeness criteria. (**A**,**B**) Structure of the SJ10 (NSC772862) small molecule, bioavailability radar (BA), displaying the six physicochemical properties of absorption including lipophilicity (XLOGP3 = 3.90), molecular weight (349.10 g/mol), polarity (PSA = 37.08 Å^2^), solubility (Log S (ESOL) = −4.7), flexibility (rotation = 4), saturation (fraction Csp3 = 0.2), and pKa of the most basic or acidic group (= 0.5) of the SJ10 compound. In addition, the SJ10 compound demonstrated a highly-probable GIA absorption, bioavailability score (55%) and good synthetic accessibility (2.89). (**C**) The compound passed the blood–brain barrier (BBB) with a score of 4.98, and further displayed a drug-like model score of −0.68. A structural characterization of the compounds was done with the help of spectroscopic studies including IR, proton NMR, 13C NMR, MS, and elemental analysis (**D**).

**Figure 4 cancers-14-00262-f004:**
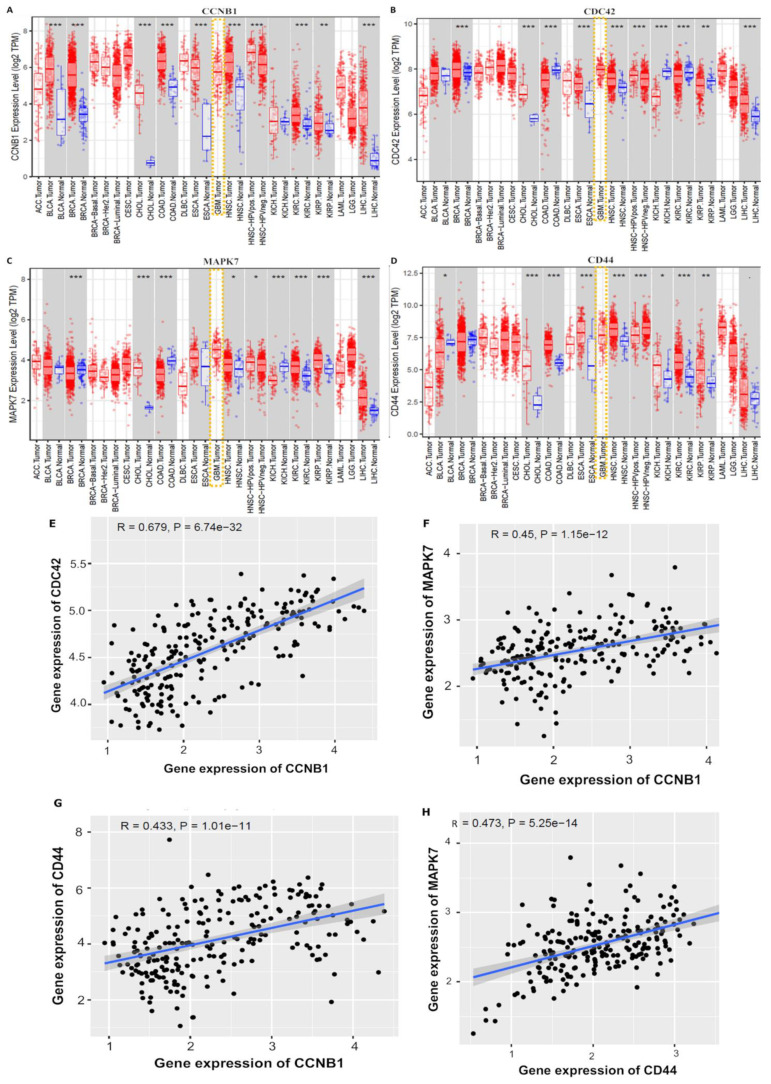
*CCNB1/CDC42/MAPK7/CD44* oncogenic signatures are overexpressed in glioblastoma multiforme (GBM). (**A**–**D**) Increased mRNA levels of *CCNB1/CDC42/MAPK7/CD44* in pan cancers, including GBM tumor tissues, compared to normal tissues from The Cancer Genome Atlas (TCGA). The relative gene expression level is indicated as transcripts per million (TPM), and expression values were normalized by log transformation of the statistical significance as evaluated by the Wilcoxon test, with *p* value significant codes: 0 ≤ ******* < 0.001 ≤ ****** < 0.01 ≤ ***** < 0.05 ≤. < 0.1. (**E**–**H**) Correlation analysis of *CCNB1/CDC42/MAPK7/CD44* oncogenes revealed correlations among all four genes when combined for analysis. Predicted results showed positive correlations ranging *r* = 0.43~0.67 of CCNB1 with CDC42, CCNB1 with MAPK7, CCNB1 with CD44, and MAPK7 with CD44 in GBM samples, with positive Pearson correlation coefficient and *p* < 0.05 considered statistically significant.

**Figure 5 cancers-14-00262-f005:**
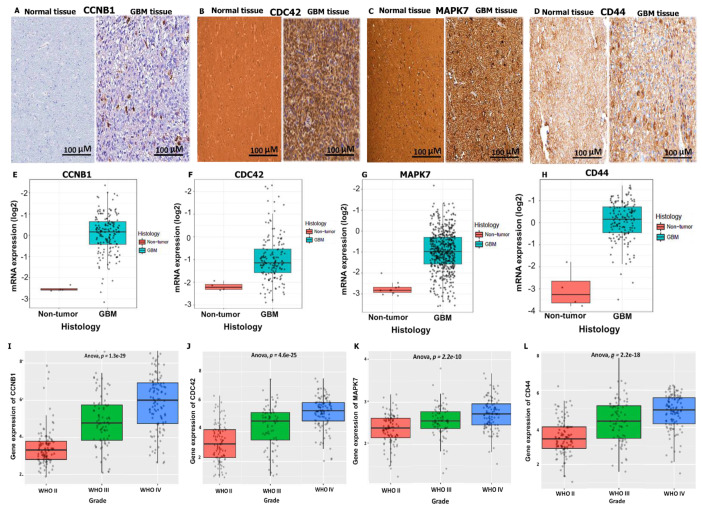
Increased expressions of *CCNB1/CDC42/MAPK7/CD44* oncogenic signatures in glioblastoma. (**A**) CCNB1 displayed medium IHC staining, with strong intensity and quantity (25%). (**B**) CDC42 displayed medium IHC staining, with moderate intensity and quantity (75%). (**C**,**D**) MAPK7 and CD44 displayed high IHC staining with strong intensity and quantity (75%), in GBM tissues as compared to normal tissues. (**E**–**H**) Increased mRNA expression levels of *CCNB1/CDC42/MAPK7/CD44* oncogenes in GBM tissues compared to non-tumor tissues from a GlioVis database analysis. (**I**–**L**) Expressions of *CCNB1/CDC42/MAPK7/CD44* gene signatures in WHO grade II, III, and IV GBM tumors using the Analysis of variance (ANOVA), with *p* < 0.05 considered statistically significant in all datasets. All images can be found online.

**Figure 6 cancers-14-00262-f006:**
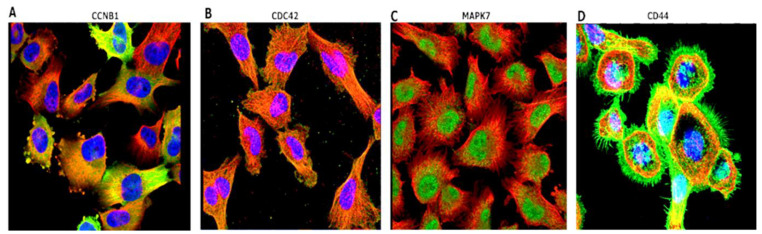
HPA staining results of the U251-MG cell line exhibited the locations of genes. Antibodies are shown in green, nuclei in blue, and microtubules in red. (**A**) CCNB1 was localized in the cytosol, (**B**) CDC42 was detected in microtubules, (**C**) while the localization of MAPK7 was in the nucleoplasm, and (**D**) CD44 was found in plasma membranes. All images are available online.

**Figure 7 cancers-14-00262-f007:**
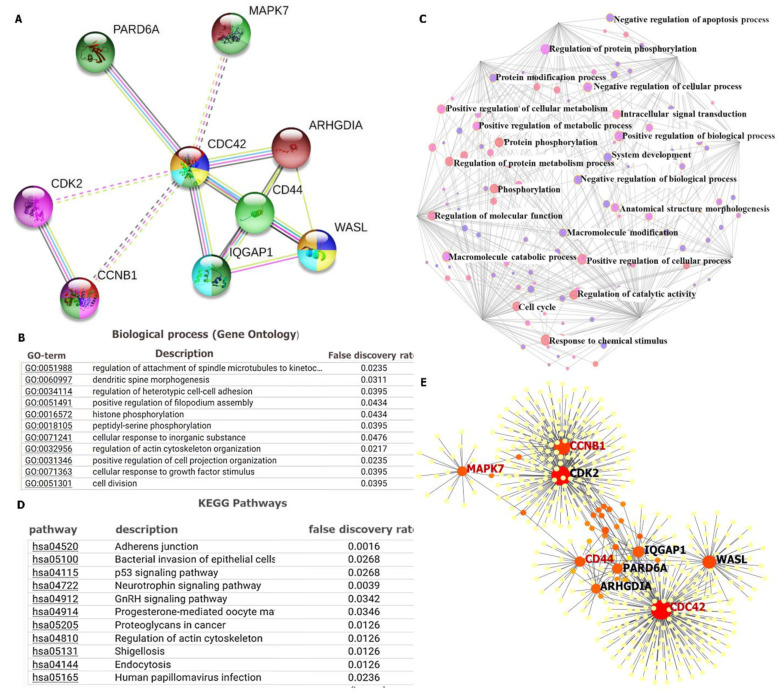
Protein–protein interaction (PPI) network revealed interactions among the *CCNB1/CDC42/MAPK7/CD44* oncogenes in glioblastoma multiforme (GBM). (**A**) The clustering network consisted of nine nodes and 15 edges, with an average local clustering coefficient of 0.917 and a PPI enrichment *p* value of 0.0293. Moreover, the interaction score confidence was set to >0.4, and *p* < 0.05 was considered statistically significant. Active interactions were based on text mining, experiments, databases, co-expressions, neighborhood, gene fusion, and co-occurrence. (**B**) The top biological processes (BPs), (**C**) KEGG pathways, and (**D**) Signaling network analysis from the BP database, showing that co-expressions of the *CCNB1/CDC42/MAPK7/CD44* oncogenes displayed enrichment in the cell cycle, regulation of molecular function, positive regulation of cellular processes, regulation of protein metabolic processes, and protein phosphorylation among others (red bubble). (**E**) Signaling network analysis of the KEGG pathway enrichment analysis showed co-expression of *CCNB1/CDC42/MAPK7/CD44* oncogenes in the same network cluster.

**Figure 8 cancers-14-00262-f008:**
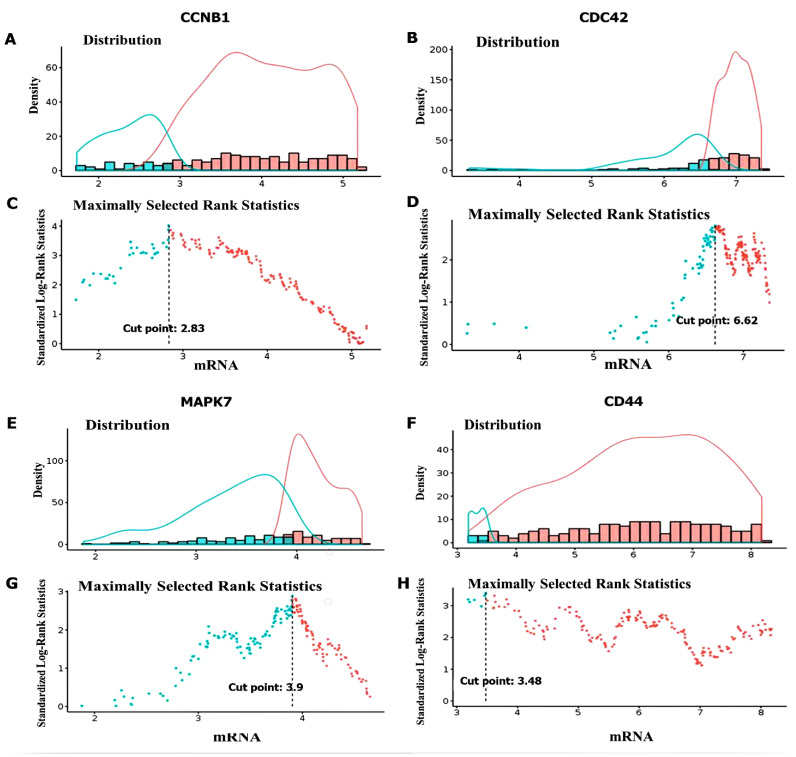
Optimal cutoff score calculations of Radscores of *CCNB1, CDC42, MAPK7,* and *CD44* expressions in glioblastoma multiforme (GBM). (**A**,**B**) Radscores plot of *CCNB1* with a cutoff value of 2.83. (**C**,**D**) Radscores plot of *CDC42* with a cutoff value of 6.62. (**E**,**F**) Radscore plot of *MAPK7* with a cutoff value of 3.9. (**G**,**H**) Radscores plot of *CD44* with a cutoff value of 3.48. The low radscores are indicated in blue and high radscores are indicated in red. This analysis shows that patients with lower Radscores generally displayed better overall survival. Therefore, predicted expressions of the *CCNB1, CDC42, MAPK7,* and *CD44* oncogenes exhibited significant roles in the cell cycle, and are thus potential prognostic biomarkers for GBM.

**Figure 9 cancers-14-00262-f009:**
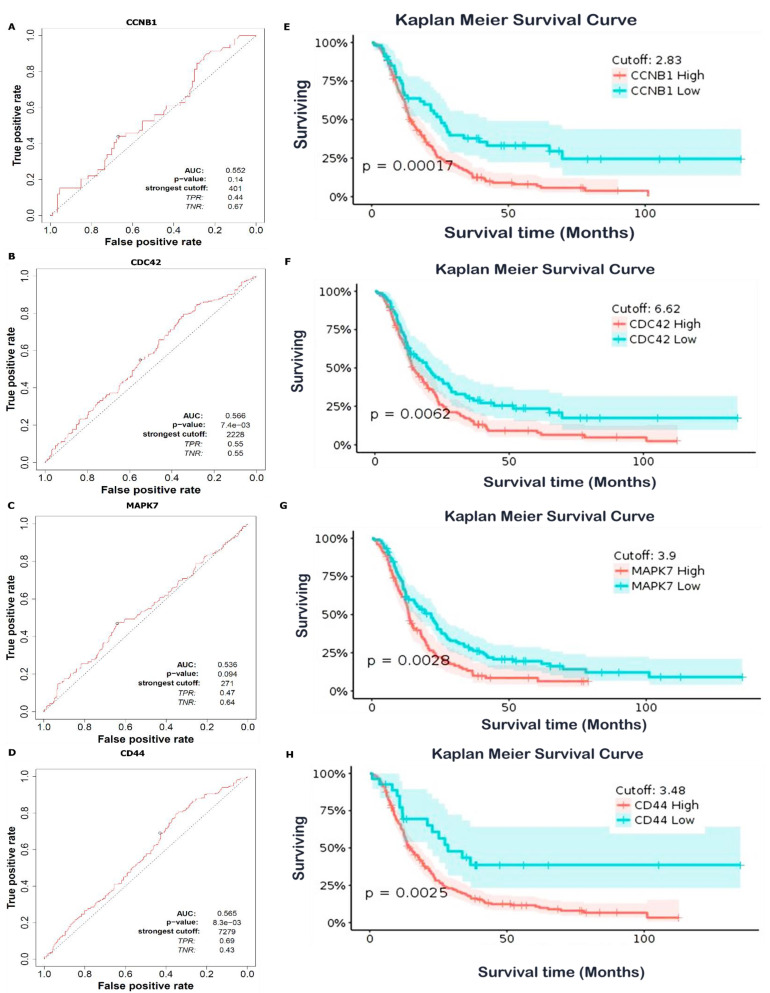
High expression levels of the *CCNB1, CDC42, MAPK7,* and *CD44* oncogenes were associated with a poor prognosis in glioblastoma multiforme (GBM). (**A**–**D**) Time-dependent ROC analysis according of the true positive (sensitivity) and false positive (specificity) rates of survival, assessed by the prognostic accuracy based on AUC values. *CCNB1* (AUC: 0.552), *CDC42* (AUC: 0.566), *MAPK7* (AUC: 0.536), and *CD44* (AUC: 0.565). An AUC of 0.5 indicates no discrimination, while an AUC of 1.0 indicates discrimination. (**E**–**H**) Kaplan–Meier analysis predicted a significant prolonged overall survival time in the low-risk group compared to the high-risk group. The analysis was based on the optimal cutoff point from the low- and high-risk groups, and *p* < 0.05 was considered significant.

**Figure 10 cancers-14-00262-f010:**
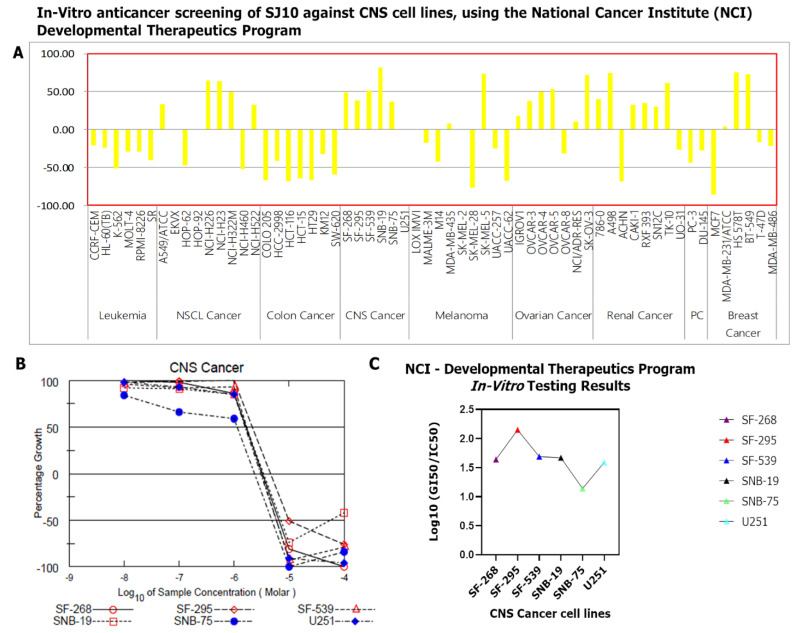
In vitro anticancer screening of SJ10 against NC1-60 CNS cell lines (**A**) The initial dose was 10 μM. Results showed that SJ10 exhibited antiproliferative activities against various CNS cell lines. The compound growth inhibition (GI) percentage showed that SNB-19 was more sensitive, with GI of 67.25%, followed by SF-539 at 36.91%, SF-268 at 34.33%, SF-295 at 23.75%, and SNB-75 at 22.34%. (**B**) Dose-dependent treatment results. SF-268 displayed complete growth inhibition at −100%, followed by U251 at −96%, SNB-75 at −84%, SF-539 at −79%, SF-295 at −76%, and SNB-19 at −42%. (**C**) SRB dual-pass staining was used to further investigate the in vitro GI_50_/IC_50_, and results ranged 1.14~2.15 μM for the CNS cell lines, with SNB-75 cells more sensitive at 1.14 μM, followed by U251 at 1.59 μM, SF-268 at 1.64 μM, SNB-19 at 1.67 μM, SF-539 at 1.69 μM, and SF-295 less responsive at 2.15 μM.

**Figure 11 cancers-14-00262-f011:**
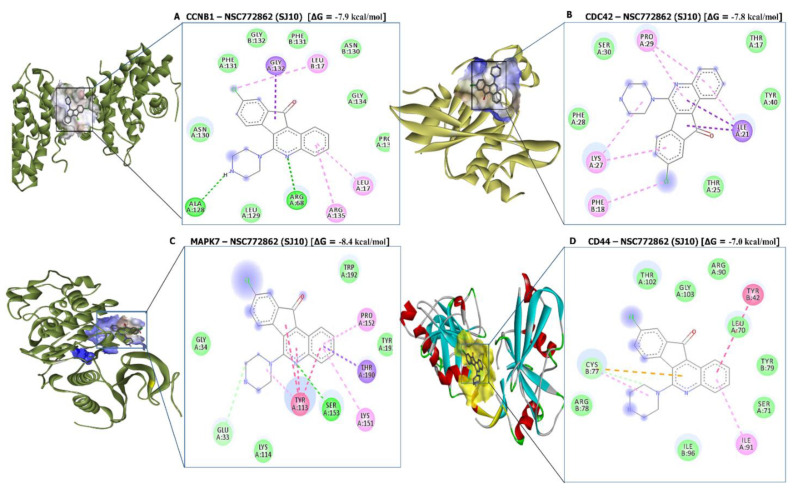
In silico docking results of SJ10 in complex with the *CCNB1, CDC42, MAPK7,* and *CD44* oncogenes in 2D representations. (**A**) The CCNB-SJ10 complex exhibited a putative binding energy of −7.9 kcal/mol, and displayed interactions by conventional H-bonds (green) with ALA128 and ARG68, and short binding distances of 2.07 and 2.73 Å, respectively. (**B**) The CDC42-SJ10 complex showed a binding energy of −7.8 kcal/mol, and displayed van der Waals interactions (THR25, PHE28, SER30, THR17, and TYR40), pi-sigma (ILE21), and pi-alkyl (PHE18, LYS27, and PRO29) in their binding pockets. (**C**) The MAPK7-SJ10 complex displayed a unique binding energy of −8.4 kcal/mol, and further showed conventional hydrogen bonds (SER153), with a shorter binding distance of 2.23 Å. (**D**) The CD44-SJ10 complex showed a binding energy of −7.8 kcal/mol, and exhibited van der Waals interactions (THR102, GLY103, ARG90, LEU70, TYR79, SER71, ILE96, and ARG78), carbon hydrogen bonds (CYS77), pi-pi T-shaped (TYR42), and pi-alkyl (ILE91) in their binding pockets.

**Figure 12 cancers-14-00262-f012:**
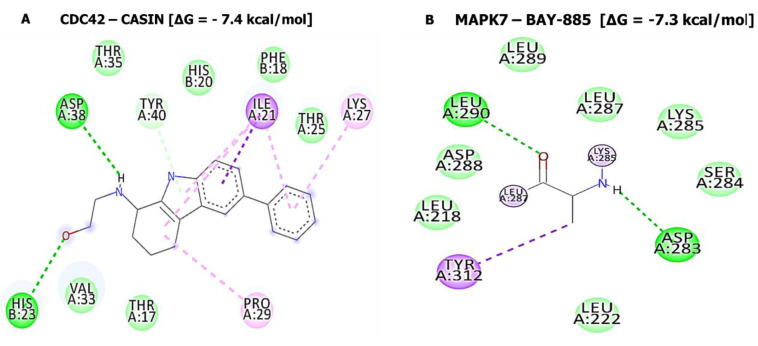
In silico docking results *CDC42* and *MAPK7* with standard inhibitors (**A**) interaction of CDC42 in complex with CASIN exhibited binding energy of (−7.4 kcal/mol). (**B**) MAPK7 in complex with BAY-885 displayed binding energy of (−7.3 kcal/mol), in 2D representations.

**Table 1 cancers-14-00262-t001:** Prediction of Biological Activity Spectra (PASS) and Swiss target genes and classes of the SJ10 compound.

SwissTarget Prediction		PASS Prediction Results		
Target Gene	Target Class	PA	PI	Activities
AKT1	Kinase	0.703	0.054	MAP kinase kinase 4 inhibitor
MAPK8	Kinase	0.652	0.014	Histidine kinase inhibitor
TTK	Kinase	0.598	0.088	Cyclic AMP phosphodiesterase inhibitor
PIM2	Kinase	0.513	0.024	MAP3K5 inhibitor
CDK9	Kinase	0.487	0.037	Antineoplastic (glioblastoma multiforme)
SLC6A3	ECT	0.529	0.091	Protein kinase inhibitor
EGFR	Kinase	0.422	0.004	Focal adhesion kinase inhibitor
CDK2	Kinase	0.445	0.040	Cyclin B1 inhibitor
MAPK7	Kinase	0.435	0.037	Apoptosis agonist
MAPK9	Kinase	0.415	0.021	Protein kinase B gamma inhibitor
CCNA2 CDK2	Kinase	0.541	0.152	MAP kinase kinase 7 inhibitor
CCND1 CDK4	Kinase	0.395	0.020	Transcription factor STAT3 inhibitor
CDK1 CCNB1	Other cytosolic protein	0.407	0.049	T cell inhibitor
CDK2 CCNA1 CCNA2	Other cytosolic protein	0.407	0.055	Wee-1 tyrosine kinase inhibitor
MAPK1	Kinase	0.353	0.042	CDC42 inhibitor
MAPK3	Kinase	0.406	0.102	Check point kinase 2 inhibitor

Pa > Pi, Pa, probability of being active; Pi, probability of being inactive.

**Table 2 cancers-14-00262-t002:** Analytical summary table showing interactions of SJ10 with *CCNB1/CDC42/MAPK7/* CD44 oncogenes.

**SJ10-CCNB1 Complex (=−7.9 kcal/mol)**		**SJ10-CDC42 Complex (=−7.6 kcal/mol)**	
Type of interactions and number of bonds	distance of interacting Amino acids	Type of interactions and number of bonds	distance of interacting Amino acids
Conventional Hydrogen bond (2)	ALA128 (2.07 Å) and ARG68 (2.73Å)	Van der Waals forces	THR25, PHE28, SER30, THR17, and TYR40
Van der Waals forces	ASN130, LEU129, PHE131, GLY132, PHE131, ASN130, GLY134, and PRO136	Pi-Sigma	ILE21
Pi-Sigma	GLY132	Pi-alkyl	PHE18, LYS27, PRO29
Pi-Alkyl	LEU17, LEU17, and ARG135		
**SJ10-MAPK7 Complex (=−8.4 kcal/mol)**		**SJ10-CD44 Complex (=−7.0 kcal/mol**)	
Type of interactions and number of bonds	distance of interacting Amino acids	Type of interactions and number of bonds	distance of interacting Amino acids
Conventional Hydrogen bond (1)	SER153 (2.23 Å)	Van der Waals forces	THR102, GLY103, ARG90, LEU70, TYR79, SER71, ILE96, and ARG78
Van der Waals forces	THR102, GLY103, ARG90, LEU70, TYR79, SER71, ILE96, and ARG78 TRP192, THR193	Carbon hydrogen bond	CYS77
Carbon hydrogen bond	CYS77	pi-pi T-shaped	TYR42
Pi-sigma	THR190	Pi-Alkyl	ILE91
Pi-Alkyl	ILE91		
Pi-Pi stacked	TYR113		
Pi-alkyl	PRO152 and LYS151		

## Data Availability

The datasets generated and/or analyzed in this study are available upon reasonable request.

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
