# Peer review of "Anticancer Activities of 9-chloro-6-(piperazin-1-yl)-11H-indeno[1,2-c] quinolin-11-one (SJ10) in Glioblastoma Multiforme (GBM) Chemoradioresistant Cell Cycle-Related Oncogenic Signatures"

_cancers, 2022, doi:10.3390/cancers14010262_

Round 1

Reviewer 1 Report

There are several comments on the submitted materials:
1. The characteristics of the test compound SJ10 and the results of the quality and purity assessment (MS, MRI and other similar methods for evaluating compounds) are not presented.
2. Experimental data required
on the effectiveness of SJ10 on several glioblastoma cell lines
3. Experimental data on the effect of SJ10 on CCNB1/CDC42/MAPK7/CD44 genes are required.
Once these comments have been eliminated, the article may be recommended for publication.

Reviewer 2 Report

In the manuscript entitled ‘Anticancer Activities of 9-chloro-6-(piperazin-1-yl)-11H-in- deno[1,2-c] quinolin-11-one (SJ10) in Glioblastoma Multiforme  (GBM) Chemoradioresistant Cell Cycle-Related Oncogenic Signatures’, Mokgautsi et al. report on increased levels of the cell division control protein homolog (CDC42), a protein which is also involved in regulating the cell cycle through the G1 phase in GBM tissues. They suggest crosstalk among CCNB1/CDC42/MAPK7/cluster of differentiation 44 (CD44) oncogenic signatures in GBM through 49 the cell cycle. They further demonstrate that a synthetic small molecule, SJ10, is a potential target agent of the CCNB1/CDC42/MAPK7/CD44 genes. They conclude that SJ10 51 target these genes and displays inhibitory activities against these oncogenes. In general, the authors provided a significant level of evidence supporting their findings in this manuscript. The manuscript is well written and presents remarkable experimental results, which may very well advance the understanding of interactions between CCNB1, CDC42, MAPK7, and CD44 proteins. However, I recommend that the following minor issues be addressed before publication.

Minor Revision

  1. Authors should review the manuscript again for minor typos e.g.
    1. Page 3 paragraph 1 line 112; correct capitalization of the word after the comma in ‘previous preliminary studies in drug discovery [38,39], We synthesized 9-chloro-6-(piper-‘
    2. Figure 5 legend, line 325; what is mulTable? ‘Figure 5. Increased expressions of CCNB1/CDC42/MAPK7/CD44 oncogenic signatures in glioblastoma mulTable 1. dis- ‘
  2. Figure 2 D-F; authors should provide a little more information on the relevance/contribution of the volcano plots to the gene expression results.
  3. Figure 6; authors should provide images that are more representative of Figure 5A and 5B, or provide images for the normal tissue samples. It is difficult to argue that the CCNB1 stain is medium and not low without showing normal tissue staining. Also, compared to CD44, it appears CDC42 staining is more intensive, so describing it as medium staining does not match well with the image presented.

Round 2

Reviewer 1 Report

-